# GPNET: Monocular 3D Vehicle Detection Based on Lightweight Wheel Grounding Point Detection Network

## Abstract

We present a method to infer 3D location and orientation of vehicles on a single image. To tackle this problem, we optimize the mapping relation between the vehicle's wheel grounding point on the image and the real location of the wheel in the 3D real world coordinate. Here we also integrate three task priors, including a ground plane constraint and vehicle wheel grounding point position, as well as a small projection error from the image to the ground plane. And a robust light network for grounding point detection in autopilot is proposed based on the vehicle and wheel detection result. In the light grounding point detection network, the DSNT key point regression method is used for balancing the speed of convergence and the accuracy of position, which has been proved more robust and accurate compared with the other key point detection methods. With more, the size of grounding point detection network is less than 1 MB, which can be executed quickly on the embedded environment. The code will be available soon.

## 1 Introduction

3D location and orientation detection is a basic but challenging problem in computer vision, which focuses on the prediction accuracy of visible and invisible points. It has been applied in many ways, including human action recognition, human-computer interaction, recently popular object detection and so on. In our application scenario, we define the point of wheel contacting with the ground as the keypoint in the vehicle instance. This paper mainly solves the problem of non-fixed number of vehicle keypoint detection, which is the basis of vehicle automatic driving perception technology. Recent researches have shown that deep convolutional network has powerful ability in information acquisition and image processing. Advanced network structures, such as Hourglass (Newell et al., 2016), HRNet (Sun et al., 2019) etc., usually have multi-scale architectures in critical point detection tasks. Location and orientation estimation tasks based on above networks with efficient transposed convolution structure can effectively solve the problem of invisible points in inference. Because they effectively combine the context information in different receptive field to ensure the high-level semantic information and high resolution information fusion at the same time. The fusion in inference process provides a rich multi-level information. This is also an important method to improve the detection accuracy of keypoints of fixed quantity.

Different from the current keypoint detection tasks, we aim to solve the problem of non-fixed number of keypoint detection. Due to the influence of shooting angle and distance, the purpose of keypoints of vehicles visible in the sample is not fixed and fluctuates violently. To adapt to the application scenes and avoid the disastrous consequences caused by the potential wrong inference of invisible points, we only forecast visible points in the image.

We adopt the top-down keypoint detection strategy and put forward a novel detection process, constraining the location information of the keypoint through wheel detection. Meanwhile, the wheel area also provides abundant pixel information for keypoint detection, including visual identifiable geometric and location information. After obtaining the grounding point information, we project the 2D point to 3D coordinate. Finally, we fuse and process multi-frame vehicle's location and orientation information to complete visual-only vehicle trajectory description.

The inference process of vehicle trajectory proposed by us has the following innovation points:

- We obtain the location and orientation in the 3D coordinate information of the vehicle by the way of keypoint detection, which doesn't need radar and 3D detection technology. It reduces the time cost greatly and ensures the real-time performance at the same time.

- We directly constrain the quantitative changes of keypoints through auxiliary target information, ensuring the continuity of updating the model gradient.

- We combine the information of multiple frames and infer the vehicle trajectory which contains location and orientation in the 3D coordinate information of the vehicle, with high information richness.

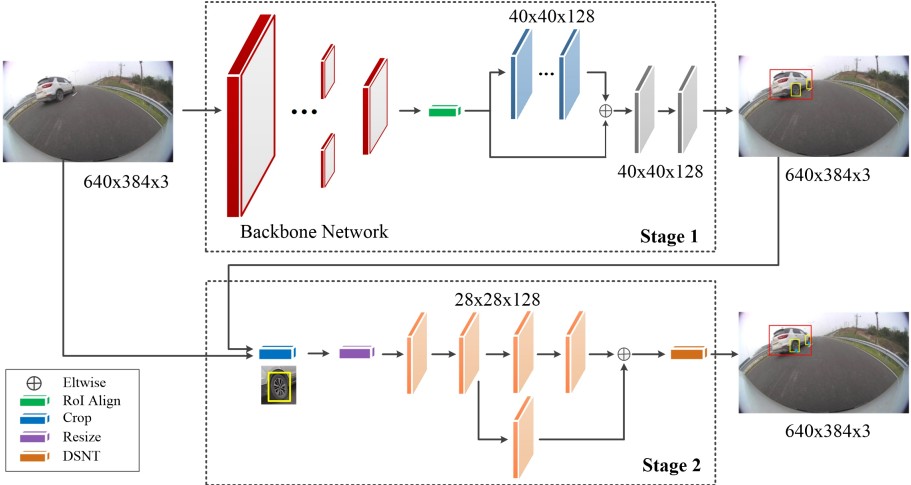

Figure 1: The structure of GPNet.

## 2 RELATED WORK

The deep convolution neural network has powerful information processing capability that makes it dominate the keypoint detection task (Belagiannis & Zisserman, 2017; Bulat & Tzimiropoulos, 2016; G. Papandreou & Murphy, 2017; Haoshu Fang, 2016; Papandreou et al., 2018; X. Sun & Wei, 2017).The current popular keypoint detection methods are usually based on the deep convolution network (Belagiannis & Zisserman, 2017; Bulat & Tzimiropoulos, 2016) . From the form of the keypoint label, there are two excellent methods: the one-hot vector (He et al., 2018) and the heatmap (F. Xia & Yuille, 2017; Girshick et al., 2014; Pfister et al., 2015; Ren et al., 2017; Kaiming He, 2015).

**One-hot mask.** In Mask R-CNN (He et al., 2018), the labels are encoded as a one-hot mask where each class of keypoint corresponds a mask. The mask is predicted by Mask R-CNN. For each of the K keypoints of an instance, the network outputs a one-hot binary mask with only one pixel is marked as the foreground.

**Heatmap regression and optimizing strategy.** The concept of heatmap is first presented by Pfister et al. (2015). The heatmap has local correlation that is similar to the feature map output by deep convolution neural network. There are many approaches proposed aiming to accelerate the network training based on heatmap, such as G-RMI (G. Papandreou & Murphy, 2017) and DSNT (Nibali et al., 2018), etc. These works divide heatmap regression problems into different sub-tasks or create techniques to decrease the regression difficulty. For instance, the heatmap is decomposed in a probability map and an offset map in G-RMI. The probability of a point being a keypoint, and the offset map describes the relative offset of the keypoint. The DSNT compresses the heatmap in size $M \times N$ that satisfies Gaussian distribution into a $2 \times 1$ vector , which decrease the difficulty of regression.

**Our Approach.** Our model is based on the top-down strategy (Chen et al., 2018; Newell et al., 2017; G. Papandreou & Murphy, 2017) in two phases. However, our approach is different from most existing works. The difculty encountered here is the number of visible keypoints varies. We utilize feature maps as the input of the second stage instead of initial image inspired by intermediate

supervision (Szegedy et al., 2015). Compared with the initial image, feature maps contain richer information which can improve the model performance of keypoint detection.

The two phase detection is shown as Figure1, we locate the grounding point in the second stage through the wheel location information obtained in the rst stage. This method takes less time/This method more efficient obtains the accurate location and orientation information in the real 3D world coordinate. Fully utilizing the geometric constraints between the vehicle and wheel ensures the gradient of the keypoints is continuous, and completely avoids the disturbance from truncate.

**GPNet.** Our approach focuses on learning the size information at the pixel level, therefore we propose a dual-branch prediction network. With the same amount of parameters, dual-branch structure can achieve higher accuracy than single-branch structure.As shown in Figure 1.

**Gradient-based OHKM.** We propose a novel online hard keypoints mining method based on the current gradients.

$$\hat{diff} = \left\{ \begin{array}{l} scale \times \hat{diff}, \\ \hat{diff}, \end{array} \right. \tag{1}$$

where $scale$ is the scaling coefficient of the gradient, $\hat{diff}$ is the initial gradient of the network, and $dt$ is the lower limit of the gradient. In this experiment, we select $dt$ as 0:05 and $scale$ as 0:1. We adjust the weights instead of truncating the gradient, which is different from OHKM(Cao et al., 2017). Our approach ensures the continuity of gradient updates while mining hard keypoints. As a result, our method has better performance, as shown in Table 1

**Fixed Range of Softmax Inputs.** Just like the principle in argsoftmax, softmax has different corresponding to different range inputs, in order to improve the performance of the proposed network, we normalize the range of input in $[0, 20]$ before softmax operation. As shown in Table2, it can greatly speeds up the training process and obtains the optimal gradient response. And experiments proved that this process can accelerate the training process and improve the performance of the proposed network

## 3 METHOD INTRODUCTION

### 3.1 TOP - DOWN STRATEGIES

Vehicle keypoint detection requires high accuracy for close vehicles. However, for the distant object only location information is required. Therefore, we follow the top-down keypoint detection strategy, which only performs keypoint detection on a single instance at a time to achieve higher accuracy.

### 3.2 WHEEL DETECTION

The network structure of wheel detection is shown in the Figure 1. The total network of keypoint detection is shown as Figure 3.

We cite the RoI Align module in mask R-CNN to cut down the pooling quantization error. And for further improving the recall rate of small targets, especially the wheel recall rate, residual structure is applied here as backbone to provide rich and fine-grained information for the model.

### 3.3 KEYPOINT DETECTION

In order to accelerate the convergence of the model and ensure the accuracy, we use DSNT to compress the spatial continuous spatial information of the network feature map. The normalized feature map of each channel maps to a two-dimensional mean point through two mapping matrices by DSNT. The mean value represents the position information of the peak point satisfying the distribution of heat map. The mapping matrices is as Equation (2) and Equation (3) separately.

$$X_{i,j} = \frac{2i - (H + 1)}{H}, i \in [1, H] \tag{2}$$

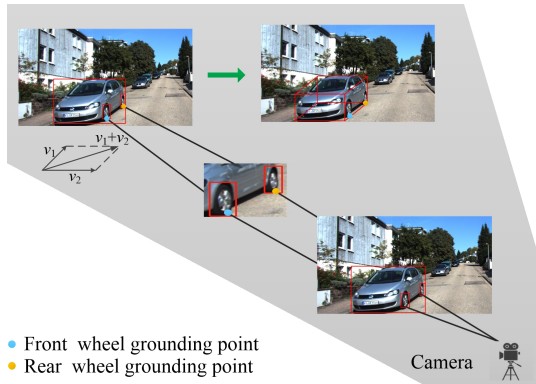

Figure 2: Wheel grounding point detection process.

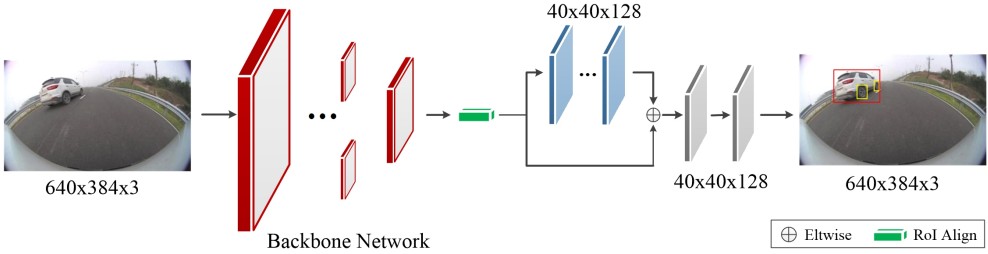

Figure 3: The first-stage of GPNET.

$$Y_{i,j} = \frac{2j - (W+1)}{W}, j \in [1, W] \tag{3}$$

where $H$ and $W$ are the width and height of the feature map separately, $X_{i,j}$ and $Y_{i,j}$ represents the value at $(i, j)$ on the matching matrix $X$ and $Y$ separately.So the mapping process is as follows:

$$\hat{x} = \left\langle \hat{Z}, X \right\rangle_F \tag{4}$$

$$\hat{y} = \left\langle \hat{Z}, Y \right\rangle_F \tag{5}$$

where $<, >$ is the sum of the two matrices after the dot product. $\hat{Z}$ represents the predicted value of the network, $X$ and $Y$ are the matrices described in Equation (2) and Equation (3). $\hat{x}, \hat{y}$ represents the horizontal index value and the vertical index value of the peak point coordinates on $\hat{Z}$. Therefore, the regression of $H \times W$ heatmap can be converted into the regression of $2 \times 1$ vector, and the transformation is spatially continuous. We also perform the same compression operation on the real heatmap.

$$x = \langle Z, X \rangle_F \tag{6}$$

$$y = \langle Z, Y \rangle_F \tag{7}$$

Where $x$, $y$ represent the values of the real heatmap Z through the matrix $X$, $Y$ compression.We choose the Euclidean distance between the label compression tensor $L_{xy}$ and the predicted compression tensor $D_{xy}$ as the loss function:

$$loss = \frac{1}{N} \sum_{y=1}^{H} \sum_{x=1}^{W} (D_{xy} - L_{xy}), N = H \times W \tag{8}$$

Where $W, H$ represent the high and width of the heatmap.

## 3.4 INFERENCE OF LOCATION

In the process of inference, the position of each point should be determined. After inference through the network, the output result is handled like Equation **??** and Equation **??**:

$$loc_{\hat{y}} = \arg\min_{i,j}(\hat{y} - Y_{ij}) \tag{9}$$

$$loc_{\hat{x}} = \arg\min_{i,j}(\hat{x} - X_{ij}) \tag{10}$$

Where $loc_{\hat{x}}$, $loc_{\hat{y}}$ represents the $x, y$ coordinate position of the peak point on the heatmap. $i, j$ represent the coordinate index, as described in Equation (**??**) and Equation (**??**).

In conclusion, our method can ensure the accuracy of vehicle yaw angle information while providing real-time performance. At the same time, it has strong robustness, especially when the model capacity is limited by storage space.

## 4 EXPERIMENT

In this section, the process and details of relevant experiments will be described.

We propose a new method of online hard keypoints mining based on the current gradients. It dynamically adjusts the learning rate in pixel and supports flexible gradient adjustments by artificial control coefficients. Our approach is similar to OHKM but keeps the continuity of the gradient while balancing the learning difficult of simple and hard keypoints. The experimental results comparison about different methods are shown in Table 1.

Table 1: Online hard keypoints mining experiment based on gradient. The gradient-based hard keypoints mining can improve the accuracy of model predictions, and this dynamic gradient adjustment mechanism will make the parameter update of the network more efficient. "–" represent without OHKM methods.

| Method | HM | AP | $AP^{50}$ | $AP^{75}$ | $AP^M$ | $AP^L$ | AR |
|---|---|---|---|---|---|---|---|
| | – | 63.0 | 85.8 | 68.9 | 58.1 | 68.5 | 68.6 |
| CMU-Pose (Cao et al., 2017) | OHKM | 41.5 | 68.8 | 47.3 | 37.3 | 51.5 | 51.5 |
| | **Ours** | **64.2** | **86.7** | **69.5** | **59.1** | **72.0** | **72.0** |
| | – | 63.6 | 86.6 | 69.3 | 59.7 | 70.4 | 69.3 |
| Pose-AE (Newell & Deng, 2016) | OHKM | 43.3 | 62.6 | 50.1 | 37.2 | 52.9 | 52.8 |
| | **Ours** | **64.1** | **85.6** | **72.4** | **59.5** | **73.0** | **72.9** |
| | – | 65.2 | 89.1 | 71.3 | 59.8 | 74.0 | 72.5 |
| Mask-RCNN (He et al., 2018) | OHKM | 43.0 | 65.1 | 47.0 | 36.7 | 53.1 | 52.8 |
| | **Ours** | **66.3** | **88.3** | **71.3** | **60.3** | **75.4** | **75.4** |
| | – | 68.8 | 89.0 | 75.5 | 64.2 | 75.9 | 74.5 |
| Person-Lab (Papandreou et al., 2018) | OHKM | 46.6 | 68.2 | 53.3 | 45.5 | 54.7 | 54.5 |
| | **Ours** | **71.6** | **89.7** | **74.1** | **69.8** | **75.3** | **75.3** |
| | – | 51.7 | 81.4 | 55.3 | 44.6 | 63.1 | 62.0 |
| DLA (Yu et al., 2018) | OHKM | 35.0 | 65.5 | 44.2 | 35.8 | 46.0 | 44.7 |
| | **Ours** | **60.1** | **86.7** | **66.8** | **54.7** | **70.5** | **70.5** |
| | – | 79.1 | 90.8 | 85.8 | 72.9 | 82.2 | 81.9 |
| HRNet-W32 (Sun et al., 2019) | OHKM | 42.2 | 66.7 | 54.8 | 40.7 | 55.2 | 55.1 |
| | **Ours** | **66.2** | **87.2** | **76.9** | **63.2** | **77.3** | **77.2** |

As shown in Table 2, it can greatly speeds up the training process and obtains the optimal gradient response. And experiments proved that this process can accelerate the training process and improve the performance of the proposed network.

In general, there are two ways to regress the grounding points. The first one is to regress the grounding points location information directly after obtaining the vehicle detection map, this method is based on the top-down method. Another one is to assume that the wheel bounding box is approximately rectangular, so it can be considered that the grounding point of a wheel is located at the center of the bottom edge of the bounding box.

Table 2: Effects with different scaling ratios. For our training data, when the scaling coefficient is 10, we can achieve the expected effect of accelerating convergence. The number of intervals was 20.

| Method | Scale | Iteration | AP | $AP^{50}$ | $AP^{75}$ | $AP^{M}$ | $AP^{L}$ | AR |
|---|---|---|---|---|---|---|---|---|
| | – | 35600 | 63.0 | 85.8 | 68.8 | 58.1 | 68.4 | 68.6 |
| | 4 | 33760 | 63.0 | 85.8 | 68.8 | 58.1 | 68.4 | 68.7 |
| CMU-Pose | 7 | 30820 | 63.0 | 85.8 | 68.9 | 58.1 | 68.4 | 68.7 |
| | **10** | **26960** | **63.0** | **85.8** | **68.9** | **58.1** | **68.5** | **68.6** |
| | 13 | 28120 | 63.0 | 85.8 | 68.8 | 58.1 | 68.4 | 68.5 |
| | – | 34960 | 63.6 | 86.6 | 69.3 | 59.7 | 70.2 | 69.2 |
| | 4 | 33740 | 63.6 | 86.6 | 69.3 | 59.7 | 70.3 | 69.2 |
| Pose-AE | 7 | 29520 | 63.6 | 86.6 | 69.3 | 59.7 | 70.3 | 69.3 |
| | **10** | **25900** | **63.6** | **86.6** | **69.3** | **59.7** | **70.4** | **69.3** |
| | 13 | 26120 | 63.6 | 86.6 | 69.3 | 59.7 | 70.2 | 69.2 |
| | – | 33040 | 65.2 | 89.1 | 71.3 | 59.7 | 73.8 | 72.4 |
| | 4 | 31460 | 65.2 | 89.1 | 71.3 | 59.7 | 73.9 | 72.4 |
| Mask-RCNN | 7 | 28120 | 65.2 | 89.1 | 71.3 | 59.8 | 74.0 | 72.4 |
| | **10** | **24540** | **65.2** | **89.1** | **71.3** | **59.8** | **74.0** | **72.5** |
| | 13 | 24960 | 65.2 | 89.1 | 71.3 | 59.7 | 73.8 | 72.3 |
| | – | 33800 | 68.8 | 89.0 | 75.5 | 64.2 | 75.7 | 74.4 |
| | 4 | 32080 | 68.8 | 89.0 | 75.5 | 64.2 | 75.8 | 74.4 |
| PersonLab | 7 | 28900 | 68.8 | 89.0 | 75.5 | 64.2 | 75.8 | 74.5 |
| | **10** | **25180** | **68.8** | **89.0** | **75.5** | **64.2** | **75.9** | **74.5** |
| | 13 | 25960 | 68.8 | 89.0 | 75.5 | 64.2 | 75.6 | 74.3 |
| | – | 46820 | 51.7 | 81.4 | 55.3 | 44.6 | 63.0 | 61.9 |
| | 4 | 45540 | 51.7 | 81.4 | 55.3 | 44.6 | 63.0 | 62.0 |
| DLA | 7 | 42180 | 51.7 | 81.4 | 55.3 | 44.6 | 63.1 | 62.0 |
| | **10** | **38800** | **51.7** | **81.4** | **55.3** | **44.6** | **63.1** | **62.0** |
| | 13 | 39000 | 51.7 | 81.4 | 55.2 | 44.6 | 63.0 | 61.9 |
| | – | 14520 | 75.3 | 90.8 | 85.7 | 72.9 | 82.1 | 81.8 |
| | 4 | 12800 | 79.1 | 90.8 | 85.7 | 72.9 | 82.2 | 81.8 |
| HRNet-W32 | 7 | 9740 | 79.1 | 90.8 | 85.8 | 72.9 | 82.2 | 81.9 |
| | **10** | **5800** | **79.1** | **90.8** | **85.8** | **72.9** | **82.2** | **81.9** |
| | 13 | 6360 | 79.0 | 90.8 | 85.8 | 72.9 | 82.1 | 81.8 |

Table 3: The experimental results of the three methods regressing the grounding point separately: direct regression to the grounding point, identify the center of bottom edge of the wheel as the grounding point, and two-stage strategy proposed by us.

| Method | Train-Method | AP | $AP^{50}$ | $AP^{75}$ | $AP^{M}$ | $AP^{L}$ |
|---|---|---|---|---|---|---|
| | CMU-Pose | 31.8 | 43.6 | 37.9 | 26.2 | 35.8 |
| | Pose-AE | 32.8 | 43.7 | 39.2 | 28.6 | 39.3 |
| Car Point | Mask-RCNN | 33.9 | 57.4 | 37.9 | 27.4 | 42.1 |
| | PersonLab | 36.4 | 56.1 | 42.3 | 32.5 | 39.8 |
| | DLA | 28.1 | 54.8 | 34.0 | 32.5 | 37.2 |
| | Center-net | 33.0 | 56.8 | 41.6 | 32.9 | 42.4 |
| Box-middle | – | 69.8 | 79.9 | 74.9 | 66.2 | 75.8 |
| **Our method** | – | **78.8** | **90.3** | **85.4** | **75.8** | **82.7** |

As shown in Table 4, when using a small model for training, the method of directly regressing to the grounding point from the vehicle cannot converge on our data set, the reason of non-convergence is that the high proportion of invisible points in the number of all points and its wide function range, so the low-capacity network cannot converge. However, it is impossible to obtain an accurate position information by directly taking the bottom edge midpoint of the wheel detection box as the grounding point. Because when the vehicle's course Angle changes, the shape of the wheel in the 2-d image will also change, and the constraint of rectangular shape is no longer true.Compared with the other two methods, the proposed method can get higher accuracy. The reason is that we do not make

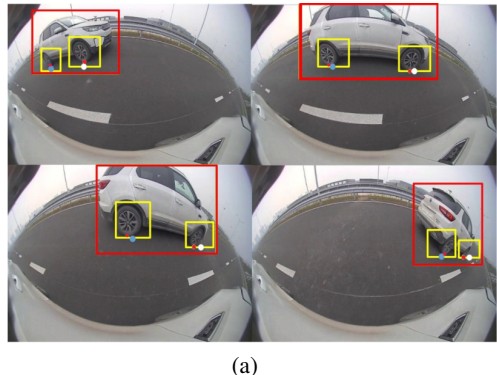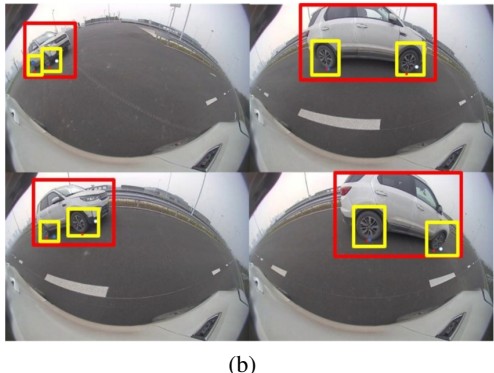

Figure 4: The red and green points are real rear-right and front-right grounding points separately, the blue and white points are the predicted results separately. (a) Identify the center of the vehicle bottom edge as the grounding point. (b) Cut off the gradients corresponding to invisible points

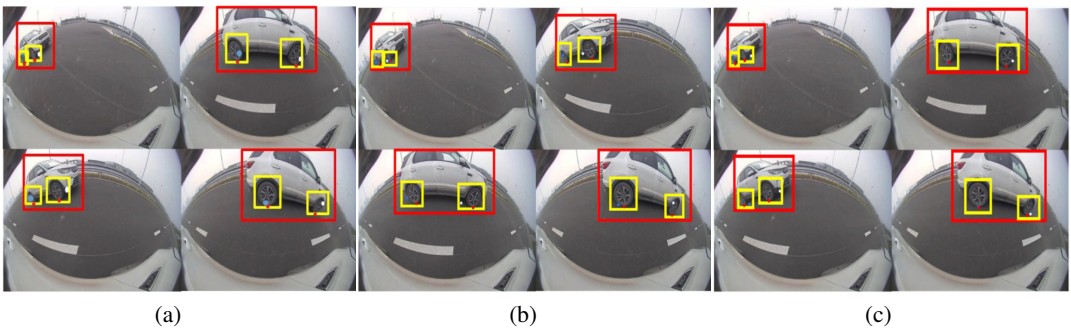

Figure 5: The red and green points are real rear-right and front-right grounding points separately, the blue and white points are the predicted results separately. (a) Invisible point at center. (b) Invisible point at top left corner. (c)Invisible point at lower right corner.

any prior constraints on the shape of the wheel and we can ensure the continuity of the gradient in training process. All that show it is necessary to detect the wheel firstly and then regress the grounding points.

In order to verify the significance of obtaining wheel position information when the proportion of invisible points fluctuate greatly and the derivative of gradient is continuous, we also conducted an experiment to fix the position of invisible points, experimental results are shown in Table 4.

Table 4: Contrast between fixing the position of invisible points and our method

| Method | Position | AP | $AP^{50}$ | $AP^{75}$ | $AP^M$ | $AP^L$ |
|---|---|---|---|---|---|---|
| | Center | 69.8 | 79.9 | 74.9 | 66.2 | 75.8 |
| Fix the position of invisible points | Left-top | 30.9 | 42.3 | 38.1 | 26.0 | 35.7 |
| | Right-down | 33.5 | 45.4 | 39.2 | 27.9 | 36.6 |
| **Our method** | – | **78.8** | **90.3** | **85.4** | **75.8** | **82.7** |

When the position of invisible points is fixed, the accuracy is 10 percent lower than the method of detecting wheel information. Comparing with the mainstream methods of truncating the gradient, it is essential to control the fluctuation of the gradient when the number of invisible points is large and the fluctuation is severe. Detecting wheel information will avoid ambiguity of target space constructed, and the proposed method can learn location information better.

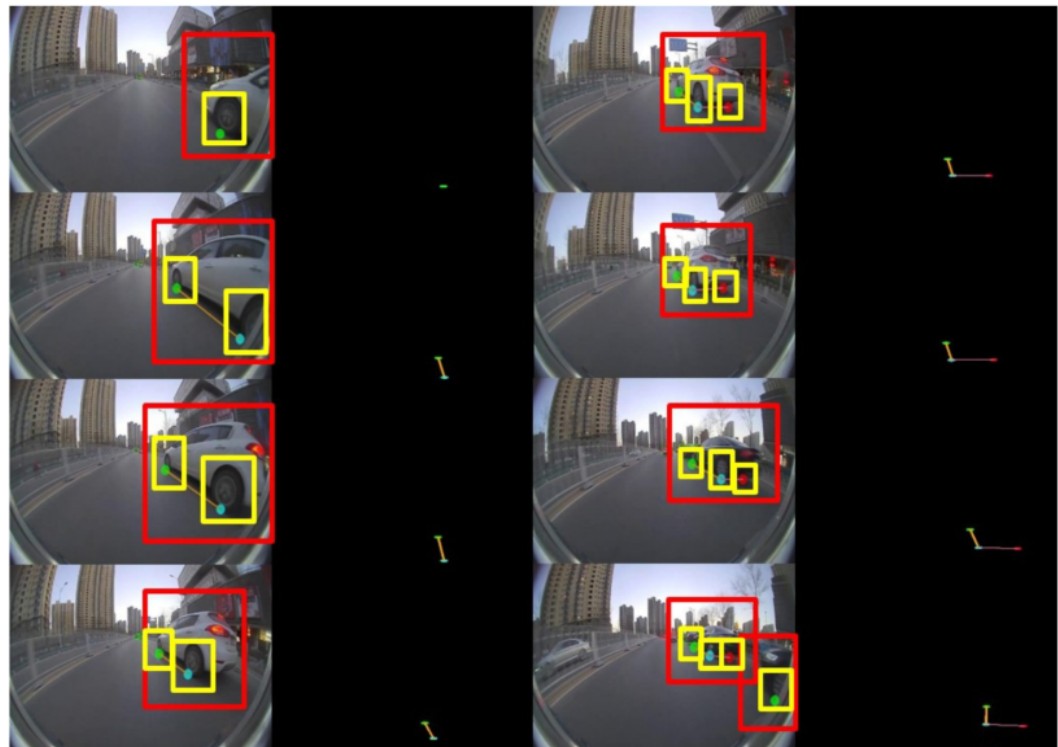

Figure 6: Keypoint detection results by our method. The cyan and creamy yellow points are real rear-left and front-left grounding points separately, the green and orange points are the predicted results separately.

As shown in Table 5, the proposed method of ground point detection is faster than the method of directly using 3D detection to obtain 3D information of the vehicle.

Table 5: Speed comparison between 3D detection and our ground keypoint dection

| Method | FPS |
|---|---|
| MV3D(Chen et al., 2016) | 2.8 |
| F-PointNet(Qi et al., 2017) | 5.9 |
| AVOD(Ku et al., 2017) | 12.5 |
| VoxelNet(Zhou & Tuzel, 2017) | 4.3 |
| ComplexYOLO(Simon et al., 2018) | 50.4 |
| **Our method** | **67.2** |

## 5 CONCLUSION

In this paper, we proposed a 3D trajectory prediction method based on a vision system. The main contribution points of the method proposed in this paper are as follows:

1) A novel 3D vehicle poses prediction method based on grounding points is proposed. This method can avoid the use of 3D detection results or Lidar information, which allows our method to work in real-time and reduce the use cost.

2) Wheel detection is adopted to avoid the drastic fluctuation of gradient updating and effectively optimize the training process.

3) In our future work, we will further combine the Re-ID and Kalman filters to obtain real3D real trajectory information for vehicle trajectory prediction.

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
