# OpenReview forum: "GPNET: MONOCULAR 3D VEHICLE DETECTION BASED ON LIGHTWEIGHT WHEEL GROUNDING POINT DETECTION NETWORK"
_ICLR.cc/2020/Conference — Reject_

### Official Review · AnonReviewer1 · 2019-10-24
**Official Blind Review #1**

**Rating:** 1

**Review:**

This paper introduces a method to detect cars from a single image. The method imposes several handcrafted constraints specific to the dataset in order to achieve higher improvement and efficiency. These constraints are quite strong and they not generalize to new situations (eg. a car in the sky, a car upside down a car with multiple wheels). The results do not seem particularly strong because the dataset seems easy and the improvements over previous works is small.

I would suggest to emphasise the improvements over previous works and send this paper to a specialized journal or venue in vehicle detection.

**Experience Assessment:**

I have published in this field for several years.

**Review Assessment: Checking Correctness Of Derivations And Theory:**

N/A

**Review Assessment: Checking Correctness Of Experiments:**

I assessed the sensibility of the experiments.

**Review Assessment: Thoroughness In Paper Reading:**

I read the paper at least twice and used my best judgement in assessing the paper.

---

### Official Review · AnonReviewer2 · 2019-10-24
**Official Blind Review #2**

**Rating:** 1

**Review:**

I find it very hard to review this paper. The idea of using keypoints to carry pose estimation is is more than 15 years old, and for the car examples reported in this paper, I'm wondering why not just you SURF or SIFT - these would certainly have been reasonable baselines. The convnets cited in this paper are mostly targeted at the harder problem of estimating human body poses.

The paper is very hard to read. It is full of typos and far from ready for submission to ICLR. The equations (eg eqn 1) are impossible to parse.

Based on all this I find it hard to trust the results.



**Experience Assessment:**

I have published in this field for several years.

**Review Assessment: Checking Correctness Of Derivations And Theory:**

I assessed the sensibility of the derivations and theory.

**Review Assessment: Checking Correctness Of Experiments:**

I assessed the sensibility of the experiments.

**Review Assessment: Thoroughness In Paper Reading:**

I read the paper at least twice and used my best judgement in assessing the paper.

---

### Official Review · AnonReviewer3 · 2019-10-29
**Official Blind Review #3**

**Rating:** 1

**Review:**

General:  The proposed method tries to improve vehicle identification and tracking by combining model-based and data-driven methods. The idea is appealing, especially the use of domain knowledge combined with a data driven method to improve a model-based task.

 The authors propose a set of model configurations on the waypoint detection, optimization techniques, a deep learning network topology and a data driven and domain knowledge based wheel detection mechanism.

Improvements on vehicle identification and tracking are not shown in the study, but different computer vision methods  are compared to each other.

The study has several shortcomings in consistently stating the problem and the result, completeness, reproducibility, quantification of results and the experimental methodology lacks  a systematic approach to isolate the effects of the different components of the proposed method.

It seems that the full method proposed yields better results compared to other way point based methods in a certain phase space. However, from the material shown, especially the lack of the experimental description and its systematic shortcomings,  I cannot judge if the components of the method can contribute to improved vehicle detection and tracking.

The paper can be strengthened by the following:
    Better framing of the problem
    Improve the descriptions on the experiments carried out
    Including a quantitative discussion on the results and the corresponding uncertainties
    Systematic studies on the behavior of the proposed method components including a mathematical description in the probability space



More detailed comments:

Better framing of the problem:

    The framing of the problem and the task lack  at least a qualitative discussion on the issues arising in using visual based vs Lidar/Radar. I disagree with the authors, that a visual system can replace these other systems, but rather enhance results under certain conditions. I am missing the discussion on the shortcomings of camera sensors with regards to the overall task: Day/Night, Fog, wheels are not visible from all angles and the impact on the target phase space. The authors give an outline, that there is a study to come tackling the performance on vehicle identification, therefore a qualitative discussion could be enough at this point.

Improve the descriptions on the experiments carried out
    So that someone else can reproduce it

Systematic studies on the behavior of the proposed method components including a mathematical description in the probability space

    The Experiment is not described at all and is therefore not reproducible. The results shown are in general the overall Precision and Recall on some dataset. The authors state at several occasions, that isolated measures on the method show better performance. E.g. the choice of the "..online hard keypoints mining method..", "..the fixed range of softmax inputs..", "..ensure the accuracy io vehicle yaw angle…". However, I cannot find proof of these statements in the material shown. Usually only the overall task performance is stated from which deductions on isolated effects cannot be drawn.
    Albeit mentioning in the introduction priors in a Gaussian Process framework, I have not noticed a notion of a random variable, a probability density distribution or the impact of the experiment on the prior pdfs.  A systematic study on the behavior of those pdfs is missing and only a few example picture of bounding boxes are shown.

Including a quantitative discussion on the results and the corresponding uncertainties
    Insert a quantitative  discussion on the experiments.  Insert estimates on uncertainties on the results or give at least a qualitative statement, if these are negligible.

Other: The paper is an application paper and does not offer novel advances for the ICLR community.


**Experience Assessment:**

I have published in this field for several years.

**Review Assessment: Checking Correctness Of Derivations And Theory:**

I assessed the sensibility of the derivations and theory.

**Review Assessment: Checking Correctness Of Experiments:**

I assessed the sensibility of the experiments.

**Review Assessment: Thoroughness In Paper Reading:**

I read the paper at least twice and used my best judgement in assessing the paper.

---

### Decision · Program_Chairs · 2019-12-19

**Decision:**

Reject

**Comment:**

This paper aims to estimate the 3D location and orientation of vehicle from a 2D image. Instead of using a CNN-based 3D detection pipeline, the authors propose to detect the vehicle’s wheel grounding points and then using the ground plane constraint for the estimation. All three reviewers provided unanimous rating of rejection. Many concerns are raised by the reviewers, including poor generalization to new situations, small improvement over prior work, low presentation quality, the lack of detailed description of the experiments, etc. The authors did not respond to the reviewers’ comments. The AC agrees with the reviewers’ comments, and recommend rejection.